# High Glucosinolate Content in Rocket Leaves (*Diplotaxis tenuifolia* and *Eruca sativa*) after Multiple Harvests Is Associated with Increased Bitterness, Pungency, and Reduced Consumer Liking

**DOI:** 10.3390/foods9121799

**Published:** 2020-12-03

**Authors:** Luke Bell, Stella Lignou, Carol Wagstaff

**Affiliations:** 1School of Agriculture, Policy & Development, University of Reading, Whiteknights, P.O. Box 237, Reading, Berkshire RG6 6AR, UK; 2School of Chemistry Food & Pharmacy, University of Reading, Whiteknights, P.O. Box 226, Reading, Berkshire RG6 6AP, UK; s.lignou@reading.ac.uk (S.L.); c.wagstaff@reading.ac.uk (C.W.)

**Keywords:** glucosinolates, rucola, arugula, *Diplotaxis*, *Eruca*, bitter taste, flavour, postharvest

## Abstract

Rocket (*Diplotaxis tenuifolia* and *Eruca sativa*) leaves delivered to the UK market are variable in appearance, taste, and flavour over the growing season. This study presents sensory and consumer analyses of rocket produce delivered to the UK over the course of one year, and evaluated the contribution of environmental and cultivation factors upon quality traits and phytochemicals called glucosinolates (GSLs). GSL abundance was positively correlated with higher average growth temperatures during the crop cycle, and perceptions of pepperiness, bitterness, and hotness. This in turn was associated with reduced liking, and corresponded to low consumer acceptance. Conversely, leaves with greater sugar content were perceived as more sweet, and had a higher correlation with consumer acceptance of the test panel. First cut leaves of rocket were favoured more by consumers, with multiple leaf cuts associated with low acceptance and higher glucosinolate concentrations. Our data suggest that the practice of harvesting rocket crops multiple times reduces consumer acceptability due to increases in GSLs, and the associated bitter, hot, and peppery perceptions some of their hydrolysis products produce. This may have significant implications for cultivation practices during seasonal transitions, where leaves typically receive multiple harvests and longer growth cycles.

## 1. Introduction

Rocket (also known as arugula and rucola) salad species such as *Diplotaxis tenuifolia* and *Eruca sativa* are leafy vegetables of the order Brassicales, and are popular throughout the world [1]. They are commonly sold in bags of loose leaves, or as part of a leafy salad mixture with other crops, such as lettuce, spinach, and watercress [2]. Previous studies have evaluated sensory properties of rocket leaves [3,4,5,6,7,8] in conjunction with phytochemical compositions, and in one instance, consumer preferences according to human taste receptor genotype [9]. One factor not accounted for in any of these studies is the temporal variability of rocket produce over the course of a growing season, and the inherent environmental variability associated with this. 

Rocket salad is in demand year-round in the UK; however, no British region is suitable for its continuous cultivation. As such, produce is typically sourced from several different countries throughout a year, according to the season [10]. In the UK, the vast majority of rocket is imported from Italy, with only seasonal summer rocket production possible in the south of England, as winters are too cold, wet, and humid for viable winter growth. In Italy, rocket is grown in the north and east during summer months; typically, in regions such as the Veneto, Lazio, and Emilia-Romagna. In winter, there is a shift in cultivation towards the south where temperatures typically remain higher, and humidity lower; for example to Campania and Apulia. In times of high demand, rocket is sourced for the UK market from other European countries (e.g., Spain), Northern Africa (e.g., Morocco), or even as far as the United States and India.

It has been well documented in the literature that crop quality and respiration rates are influenced by seasonality [11]. This is intrinsically linked to growing temperature, as metabolic rates tend to be higher under warmer conditions. Growth temperature therefore plays a distinct role in determining shelf life longevity and visual acceptability of leaves [12]. There is also evidence suggesting that pungency is increased during spring and summer months [13], though this has not been shown quantitatively in rocket. Conversely, sugars have been shown to be reduced in some Brassicales species under “high” growth temperatures (>20 °C) [14], which has implications for sensory traits. Anecdotal evidence suggests that source and season significantly affects the quality and consistency of rocket leaves. This presents a problem for producers and supermarkets as leaves are typically marketed as generic products, but the quality of the produce is not consistent.

A previous study by Bell et al. [9] highlighted that pungency (or hotness) of rocket leaves is the main driver of rocket liking. Excessive pungency is typically rejected by consumers, and most people in that study preferred milder and sweeter leaves. There was a significant plant genotypic component determining the pungency of leaves, but it remains unstudied how pungency and consumer preference may be affected by changes in climate and seasonal growth. The pungency of rocket leaves is due to the presence of glucosinolates (GSLs) within tissues, which are hydrolysed by myrosinase enzymes to produce isothiocyanates (ITCs) and numerous other products. One compound present in both *E. sativa* and *D. tenuifolia* is glucosativin (4-mercaptobutyl GSL; GSV) and its dimer (dimeric 4-mercaptobutyl GSL; DMB). These produce the ITC, 4-mercaptobutyl ITC, which undergoes spontaneous tautomeric rearrangement post-hydrolysis to form 1,3-thiazepane-2-thione (sativin; SAT) [15]. It is unknown if it is the ITC or tautomer that is responsible for pungency and flavour, but olfactometry has shown a distinctive rocket-like aroma associated with the hydrolysis product of GSV [16]. The inherent variability of GSL biosynthesis in response to growth environment affects the presence and abundance of GSV and health related GSLs (such as glucoraphanin, GRA; and glucoerucin, GER) [17], and therefore may impact upon sensory properties of leaves and consumer acceptance.

An often-neglected component of taste and flavour perception in vegetables is sugar content and composition. Sugar content of Brassicales leaves is known to be variable according to growth conditions [18]. In combination with effects on GSL composition, sugars may therefore have a strong influence upon sensory traits and consumer preference throughout a growing season.

We present phytochemical, sensory analysis, and consumer preference and perception data from a year-long study of rocket produce sourced from commercial farms delivered to UK-based processors and supermarkets. We hypothesised that GSL and sugar content would significantly affect sensory properties of rocket leaves at different times of the year. This in turn would affect consumer perceptions of leaves and their preference of rocket would change seasonally. We also present the effects of climatic factors (such as temperature) and cultivation practices imposed on rocket crops (such as multiple harvests/cuts of the same crop).

## 2. Materials and Methods 

### 2.1. Plant Material

Rocket leaf material was sourced and delivered to the University of Reading Sensory Science Centre monthly (with the exceptions of May and December) for one year (2014) by Bakkavor Ltd. (Spalding, UK). This material corresponded to leaf material delivered to a processing facility in England, and the material that would be used in products destined for UK supermarkets and consumers. Each sample batch was harvested, washed, and processed in accordance with industry practice. For reasons of commercial sensitivity, the exact locations of growers that supplied material for this study will not be named. Only the country of origin will be detailed, along with the growth environment (polytunnel, glasshouse, or open field) and cultivar used. It should be noted that as with many other crops, cultivars are selected by growers according to season and performance under specific conditions and climates. As such, the cultivars tested each month were not always the same. This was intentional to properly evaluate the level of consistency of produce as it exists within the UK supply chain, not to evaluate the changes of any single cultivar over the year or between different countries. Data were provided by growers relating to the length of each crop cycle (i.e., how long each of the samples had been grown for), the number of cuts each sample had undergone, and the percentage of dry matter of each batch at intake. See Table 1 for a summary of samples tested in each respective month.

### 2.2. Temperature Data

Temperature data for each of the rocket growing sites were supplied by Bakkavor Ltd. Data were logged throughout the cropping cycle of plants via on-farm weather stations. Four measurements were provided: the average daily temperature for the entire cropping cycle (referred to as “avg. temp”), the maximum temperature of the week preceding harvest (referred to as “max. temp. week”), the minimum temperature of the week preceding harvest (referred to as “min. temp. week”), and the average temperature in the week preceding harvest (referred to as “avg. temp. week”).

### 2.3. Sensory Analysis

A panel of 12 previously trained and experienced assessors (ten female, two male) evaluated the samples at the Sensory Science Centre (University of Reading, Reading, UK). All training and monitoring of the sensory panel was in accordance with ISO 8586:2012 and ISO 11132:2012 standards. The panellists used the consensus vocabulary developed by Bell et al. [4] to describe the samples; the developed terms included appearance, odour, taste, flavour, mouthfeel, and after-effects attributes. Panellists were familiarized with the evaluated sensory attributes prior to each monthly scoring session, with reference standards when required.

Each month the panellists rated the samples individually in isolated well ventilated, temperature-controlled booths (22 °C), under artificial daylight. Attribute intensity was scored on 15 cm unstructured line scales (data scaled 0–100). Samples were presented monadically in a balanced presentation order, coded with three-digit random codes. Cold water and natural unflavoured yoghurt were provided for palate cleansing, and warm water for washing fingers between samples. Data were collected in duplicate using Compusense (version 5.5, Guelph, ON, Canada).

### 2.4. Consumer Analysis

Consumer recruitment and assessments were conducted as per the protocols of Bell et al. [9]. Evaluations were held on a bimonthly basis. Briefly, consumers were recruited from in and around the University of Reading and asked to attend as many of the evaluation sessions as possible over the course of the year-long study. A total of 55 consumers (out of 101) attended every session of the study.

Volunteers were presented with three leaves and asked to score their liking of leaf appearance, taste, and “overall” liking of each sample on a line scale (0–10), and based on their own individual experience of rocket sensory attributes. Volunteers were also asked to score their perceptions of bitterness, hotness (pungency), sweetness, and pepperiness. Scores were entered into a general labelled magnitude scale (gLMS) ranging from “not detectable”, “weak”, “moderate”, “strong”, “very strong”, to “strongest imaginable”. Data were subsequently converted to antilog values and normalized for statistical analysis [19]. Samples were presented monadically in a randomized, balanced presentation order, with three-digit random codes in duplicate. Data were collected using Compusense (version 5.5, Guelph, ON, Canada).

The demographics and characteristics of each respective panel are presented in Appendix A. The average number of recruits for bimonthly evaluations was 87, with an average age of 35 years old, ranging from 18 to 70. Volunteers were predominantly female (70.7%, on average), which is partly due to the gender balance present within the School of Chemistry, Food and Pharmacy at the University of Reading. We acknowledge that the sample population of consumers may not be representative of the “typical” UK consumer, however it does incorporate a broad range of culturally and ethnically diverse individuals that encompass a wide diversity of potential sensory genotypes.

On average, 50.9% were employed, and 46.9% were students. Of these, 27.2% were food and nutrition students from within the school. The multicultural nature of the staff and student body produced a diverse cohort, with 48.2% identifying as “white”, 27.7% as “other” (i.e., non-white and/or European), and 6.7% of Chinese nationality. The remainder were composed of those regarding themselves as African (4.8%), Caribbean (2.3%), Indian (2.7%), or of mixed race (1%). Of the volunteers 3.3% declined to provide a response.

### 2.5. Phytochemical Analyses

#### 2.5.1. Preparation of Samples

Upon receipt of samples at the University of Reading, a subset of leaves (50 g) was taken for analysis, frozen at −80 °C, and lyophilized prior to extraction. Tissues were then milled into a fine powder using a Wiley Mini Mill (Thomas Scientific, Swedesboro, NJ, USA).

#### 2.5.2. Glucosinolate Analysis

DMB, GER, GRA, and GSV concentrations were determined by Liquid Chromatography Mass Spectrometry (LC–MS) as per the methodology presented by Bell et al. [20]. Three separate biological replicate extractions were performed on each sample, with three technical replicates analysed by LC–MS (*n* = 9). The data for each sample were then averaged to give a representative concentration of each assessment month. Individual sample averages were retained and used for subsequent PCA. Individual cultivar results for each respective month of the study can be found in Appendix A.

#### 2.5.3. Sugar Analysis

Concentrations of fructose, galactose, and glucose were determined by extraction and analysis by capillary electrophoresis (CE) according to the methodology presented by Bell et al. [4]. The same level of replication as for the analysis of GSLs was used for each sample (*n* = 9) and averaged to produce a representative monthly concentration for presentation. As above, individual sample averages were retained and used for subsequent PCA. Individual cultivar results for each respective month of the study can be found in Appendix A.

### 2.6. Statistical Analysis

#### 2.6.1. Panellist Performance

Data were collated and panellist performance evaluated using SenPAQ (v5.01; Qi Statistics, Reading, UK). For each monthly assessment, scores were averaged and used for further statistical analysis. Discrimination, repeatability, and consistency were checked for all assessors.

#### 2.6.2. Analysis of Variance

Shapiro–Wilk normality tests were conducted for all sensory and consumer variables. All of which were concluded to fit with a normal distribution and allow for statistical comparison using a parametric test. Analysis of variance (ANOVA) was performed on each data set (sensory, consumer, and phytochemical) and supplied temperature data. Each test was performed using XLSTAT (Addinsoft, Paris, France) with a protected post-hoc Tukey’s honest significant difference (HSD) test (*p*-values ≤ 0.05). Only attributes with statistically significant differences were selected for presentation.

#### 2.6.3. Agglomerative Hierarchical Clustering

Agglomerative hierarchical clustering (AHC) was conducted on the consumer liking data using XLSTAT. This approach was used to cluster consumers who had similar liking patterns (for taste and overall liking) for each of the bimonthly panels. Dissimilarity of responses was determined by Euclidean distance, and agglomeration using Ward’s method (set to automatic truncation).

#### 2.6.4. Principal Component and Correlation Analysis

Consumer liking and perception response data were used to extract principal components (PCs; with Varimax rotation) and we performed correlation analyses (Pearson, *n* − 1). Phytochemical, temperature, and agronomic data for each sample were regressed as supplementary variables within the PCA model. Variables such as month, cultivar (variety), cut, and country of origin were regressed as qualitative variables to generate categorical centroids within the model. Seven PCs were extracted with the first four components containing a cumulative 98.3% of variability. PCs 1 and 4 had eigenvalues of 4.1 and 0.3, respectively) and were selected for presentation after Varimax rotation. Correlation matrices of all attributes used in the analysis were produced at the 5%, 1%, and 0.1% significance levels, and are summarized in Appendix A.

## 3. Results and Discussion

### 3.1. Monthly Differences in Rocket Agronomic Practices

The majority of rocket supplied to the UK market is *D. tenuifolia*, with *E. sativa* making up a small amount. The latter is usually supplied in winter months due to its faster establishment, early vigour, and cold tolerance [10]. In this study *E. sativa* was only supplied in November (Table 1).

Cultivation practices varied distinctly between countries, and indeed between individual growers, based on local cultural practices and individual experience. In Italy, produce destined for the UK market is typically cultivated under polytunnel or glass, year-round; whereas UK grown material is either grown in open field or under glass (Table 1). One dominant reason for this difference is that the wetter and more humid climate of the UK can cause severe fungal pathogen outbreaks. The reduced airflow within polytunnels typically exacerbates this problem, and so open field is preferred to minimize losses.

The length of crop cycles depends on the season, though there are large differences between individual growers and countries (Table 1). Cycle length is longer in the winter and spring months, with much faster growth and regrowth in summer and autumn. The shortest average crop cycle in this study was 27 days (August), and the longest 96 (March). The extremes of the overall range (Table 1) can vary from 23 (June) to 180 days (April).

As establishment of rocket crops is more difficult in winter months, Italian growers favour repeated harvests until warmer weather arrives. It is not unusual for >5 cuts to be taken from a single sowing. During the experiment, sourced material came not only from Italy, but the USA and India (Table 1) in order to meet shortfalls in demand. During the summer season, UK rocket enters the market and typically has short growth cycles and receives only one cut. The humid climate does not favour regrowth, as damaged leaves become infected with fungal pathogens and are unsaleable.

The length of crop cycle and cut number have important implications for rocket taste, flavour, and acceptability. It is widely acknowledged that the more harvests a rocket crop undergoes, the more pungent and aromatic it becomes, due to the initiation of wound response and increases in secondary metabolites, such as GSLs [21]. However, no quantitative research has been conducted to evaluate consumer preferences for first, second, or multiple cut leaf material of rocket. As will be discussed in the following sections, cut number is a key determinant of taste and flavour perception, and liking of leaves at different times of the year.

### 3.2. Monthly Variation in Rocket Growth Temperature

Due to the seasonal distribution of rocket production geographically throughout a growing season, crops may be exposed to a range of temperature maxima and minima. Appendix A presents an average of the temperatures recorded at each farm location, giving a representative value of all growing sites for each month.

The highest average temperature across the growing season was 21 °C in August, with the highest average temperature in the week preceding harvest being 21.6 °C. The highest average maximum and minimum temperatures in the week preceding harvest were also in August; 27 °C and 15.6 °C, respectively. Lowest average temperatures were observed in January (avg. temp. 10 °C, max. temp. week 14.8 °C, min. temp. week 4.6 °C, and avg. temp week 9.8 °C). The significant differences observed in monthly temperatures correspond to distinct changes in phytochemical content, sensory perceptions, and consumer acceptance.

### 3.3. Phytochemical Composition and Monthly Variability

#### 3.3.1. Glucosinolates

The monthly average GSL concentrations of rocket leaves are presented in Figure 1a. For individual cultivar concentrations, see Appendix A. The data show a very large amount of variability over the course of the year. This lack of consistency likely plays a significant role in the perceived quality changes in rocket produce by processors, supermarkets and consumers.

Figure 1c details the significant differences between each growing month for GSL composition. Total concentrations of the four major GSLs of rocket were highest in October (17.6 ± 0.6 mg g^−1^ dw) and lowest in July (5.8 ± 0.7 mg g^−1^ dw). GRA concentrations were significantly higher in October (4.2 ± 0.1 mg g^−1^ dw) and February (3.3 ± 1.3 mg g^−1^ dw) compared with the months from March to September. GER concentrations were significantly higher in January (2.3 ± 0.1 mg g^−1^ dw) than at any other time of the study year. Previous studies in broccoli sprouts [22] have shown that cooler temperatures (<16 °C) increase the concentrations of methylthioalkyl GSLs, such as GRA and GER. This may be due to a abiotic stress response and upregulation of secondary metabolite biosynthesis, causing greater concentrations of these health related GSLs. GER, GRA, and their respective hydrolysis products are not known to have any significant odour or flavour, but the elevations observed in winter months suggests that cultivation in lower-temperature climates may improve rocket nutritional potential. Both sulforaphane (SF) and erucin (ERU; isothiocyanate hydrolysis products of GRA and GER, respectively) are known to be effective against some forms of cancer [23,24].

Concentrations of DMB were also significantly higher in January (2.6 ± 0.2 mg g^−1^ dw), October (4.2 ± 0.2 mg g^−1^ dw), and November (3.4 ± 0.4 mg g^−1^ dw), whereas amounts of the monomer GSV were highest in September (10.4 ± 1.0 mg g^−1^ dw). The relationship between GSV, DMB, and sensory properties is not understood, but previous studies have noted associations between GSV content and pungency (likely due to hydrolysis producing SAT), but not DMB [4]. The significant variations in monomer and dimer forms across the year suggest that there is some as-yet-unknown mechanism by which the two are interconverted [15]; possibly on a genetic and enzymatic level. This process may dictate the levels of pungency found in leaves.

#### 3.3.2. Sugars

The pattern of sugar accumulation in rocket leaves was much more distinct than for GSLs. Total concentrations were significantly higher from June to September (Figure 1b,c) indicating a strong relationship with seasonal climate. This could conceivably be linked to temperature and light intensity duration and quality during summer months. A study on broccoli [25] previously observed that glucose and fructose concentrations were significantly elevated under higher temperature conditions, for example.

Glucose was the dominant monosaccharide in rocket leaves, and concentrations were significantly higher from June to September (peaking in July, 93.9 ± 3.0 mg g^−1^ dw). Fructose accumulations also followed this pattern, with 27.5 ± 0.9 mg g^−1^ dw in July, compared to only 2.5 ± 0.2 mg g^−1^ dw in January. These data are strong evidence for the role of season and climate in the generation of sugars in rocket leaves; and as will be discussed, this has implications for preference and quality of leaves. 

### 3.4. Sensory Profiling Monthly Variability

#### 3.4.1. Appearance Traits

Leaf size and uniformity of size were the only two appearance attributes tested that varied significantly between monthly assessments of rocket produce (*p* = 0.005 and <0.0001, respectively; Appendix A and Table 2). Leaf size was significantly smaller in January compared with April, June, August, and October. Similarly uniformity of size was significantly lower in January than any other month (with the exception of February). Combined with the low average temperatures (Appendix A) at this time of year, it is likely that the colder temperatures and reduced light levels (short days) in Italy at this time of year result in slower, and more uneven growth rates [26] compared to other times of the year.

#### 3.4.2. Odour Traits

The odour attributes of rocket leaves defined as green, stalky, earthy, peppery, sweet, and mustard were all found to vary significantly between assessment months (Appendix A and Table 2). Green, peppery, and earthy odours were observed to be elevated, on average, in January, whereas stalky and sweet odours were scored higher in July, August, and September. Volatile profiles are known to be influenced by seasonal variations, and storage conditions [27], and so differences between the UK and Italian climates likely play a role in determining the intensity of these odours.

#### 3.4.3. Taste and Flavour Traits

Sour taste, savoury taste, stalky flavour, peppery flavour, and earthy flavour of rocket leaves were found to vary significantly between months (Appendix A and Table 2). Sour and savoury taste scores were highly variable between months, with no distinct pattern emerging according to seasonality. As with aroma attributes, stalky, peppery, and earthy flavours were each scored highest in September and January.

#### 3.4.4. Mouthfeel Traits

Significant variation was observed between monthly assessments of rocket for crisp and drying mouthfeels. Leaves tested in January were significantly less crispy than those received from March to October (Appendix A and Table 2). Soluble sugars are known to help maintain turgidity of leaves [28], and the low concentrations accumulated at this time of year may therefore be related to mouthfeel quality.

Drying sensation was perceived as significantly more in September and October than the months from March to June, and November. Little is known about the cause of drying sensation caused by rocket leaves, but one possible explanation is the presence of polyphenols [29], which have been observed to increase significantly under heat stress conditions [30].

#### 3.4.5. Aftereffect Traits

Aftereffect attributes with significant monthly variation are presented in Appendix A and Table 2. Of note are sweet and peppery aftereffects, which have previously been associated with improved consumer acceptance [9]. Sweet aftereffects were significantly higher in July, corresponding to the peak of glucose and fructose concentrations within leaves.

Peppery aftereffects were significantly higher in January, in agreement with the aroma and flavour scores for this attribute. Some GSL hydrolysis products are known to have different aromas at different concentrations [31], and the low abundances of GSV in January (Figure 1a) would suggest that SAT production may also be reduced, and correspond to reduced pungency and increased pepperiness.

### 3.5. Correlation Analysis of Sensory Attributes

#### 3.5.1. Growing Temperature

Correlation analyses and significances are presented in Appendix A. Average crop cycle temperature, the minimum, and average temperatures in the week preceding harvest were significantly correlated with sweet odour of leaves (all *r* = >0.462; *p* = <0.0001). Abiotic stress is known to promote formation of secondary metabolites in many plant species [32] and so higher growth temperatures may promote the synthesis of aldehydes that impart sweet odour, as have been identified in other Brassicales species [33].

All temperature data were also significantly correlated with crisp mouthfeel (*r* = 0.527; *p* = <0.0001). Previous research and modelling of rapeseed plants has shown that growth temperature significantly impacts leaf morphology; particularly leaf length and thickness [34]. This may partly explain why rocket leaves are perceived as crispier in summer and autumn months compared with winter (Appendix A).

#### 3.5.2. Cultivation Practice

One of the largest differences observed between months was the length of the crop cycle (Table 1). Correlation analysis (Appendix A) found that the length of the crop cycle was significantly associated with key sensory traits potentially linked with consumer acceptance. These were: bitter taste (*r* = 0.3; *p* = 0.043) and bitter aftereffects (*r* = 0.325; *p* = 0.027). Sweet aftereffects were also significantly and negatively correlated with the length of crop cycle (*r* = −0.316; *p* = 0.032). The low sugar:GSL ratio in samples with longer crop cycles might explain some of these correlations. With lower sugar concentrations in the winter/early spring months (Figure 1b), GSLs and their hydrolysis products may be perceived more strongly with the masking effect of sugars reduced. 

#### 3.5.3. Glucosinolates

Individual GSL concentrations are known to be associated with sensory attributes of rocket species [3]. GRA is a compound not known to impart any taste or flavour [31], but correlation analysis revealed significant negative associations with sweet odour (*r* = −0.543; *p* = <0.0001), taste (*r* = −0.402; *p* = 0.005), and aftereffects (*r* = −0.304; *p* = 0.035). The abundance of GRA was negatively correlated with the average growth temperature (*r* = −0.344; *p* = 0.017) and max. temperature in the week preceding harvest (*r* = −0.306; *p* = 0.035). These two points indicate that GRA biosynthesis is lower in samples grown in months with higher temperatures, which also corresponds to increased sugar concentrations (Figure 1).

GER is similar to GRA in the respect that it is not known to impart taste [31], however its hydrolysis product erucin (ERU) has been described as having a “radish-like” aroma [16]. In this study, several previously unobserved associations were found. GER itself is negatively correlated with pungent odour (*r* = −0.299; *p* = 0.039), but positively with green and peppery odours (*r* = 0.459; *p* = 0.001, and *r* = 0.364; *p* = 0.011, respectively) and flavours (*r* = 0.367; *p* = 0.01, and *r* = 0.337; *p* = 0.019, respectively). While these data are not conclusive of a causative relationship with these attributes, it does suggest that occurrence of GER in high concentrations may elicit, or be associated with, perceptions of pepperiness and green attributes, and is worth studying in greater detail in future studies.

GSV exists in a monomer and dimer form (DMB), and typically makes up the largest proportion of the GSL profile of rocket [20]. A previous study found that its hydrolysis product SAT has a “rocket-like” aroma [16]. While this may be considered a somewhat subjective description, it is speculated that SAT is responsible for the perceived pungency of rocket leaves. The data in this study agreed with this hypothesis, as GSV concentrations were significantly correlated with pungent aroma (*r* = 0.393; *p* = 0.006). It was however also negatively correlated with peppery odour (*r* = −0.295; *p* = 0.042), suggesting that the two attributes are separate, with only GSV being indirectly responsible for pungency.

Correlations of DMB with sensory attributes were distinct and separate from the monomer, suggesting that concentrations of the two forms are influenced by the environment and as-yet-unknown genetic regulation, possibly in response to abiotic stress. It is unknown if DMB itself imparts taste or flavour, but its abundance was positively correlated with savoury taste (*r* = 0.323; *p* = 0.025) and aftereffects (*r* = 0.391; *p* = 0.006). This is in agreement with previous sensory and consumer studies of rocket [4,9]. It was also observed that GSV was significantly correlated with each of the four temperature measurements used in the analysis (Appendix A; Appendix A) whereas DMB was negatively correlated with the max. temperature in the week preceding harvest (*r* = −0.311; *p* = 0.031). This suggests that the relative abundances of the monomer and dimer forms of GSV had an environmental component, with greater concentrations of GSV present in hotter months.

#### 3.5.4. Sugars

Total sugars, fructose, and glucose concentrations were significantly correlated with dry matter percentage (*r* = 0.494; *p* = 0.001, *r* = 0.622; *p* = <0.0001, and *r* = 0.439; *p* = 0.003, respectively). This suggests that this physical property of leaves may be indicative of a dry matter concentration effect. This is reflected in several negative correlations with moistness mouthfeel (*r* = −0.385; *p* = 0.007, *r* = −0.515; *p* = 0.000, and *r* = −0.332; *p* = 0.021, respectively). Only galactose concentrations were significantly correlated with sweet taste (*r* = 0.39; *p* = 0.006), and fructose and galactose with sweet aftereffects (*r* = 0.303; *p* = 0.036, and *r* = 0.308; *p* = 0.033, respectively). Despite the significantly higher sugar concentrations in summer months (Figure 1b,c) there were no significant correlations with growth temperature.

The sugar:GSL ratio was also similarly correlated with the aforementioned mouthfeel effects (Appendix A) and dry matter content (*r* = 0.571; *p* = <0.0001); but only sweet aftereffects (*r* = 0.393; *p* = 0.006) and not sweet taste. This association is not as strong as found in previous studies of rocket [4].

### 3.6. Consumer Acceptability and Perception

#### 3.6.1. Liking of Taste

Consumer liking of taste and the results of AHC are presented in Appendix A. Liking of taste is defined as liking associated with taste and flavour attributes alone (bitterness, sweetness, pepperiness, and hotness), irrespective of appearance traits. Three clusters were identified in each respective month, except for March, where four clusters were observed. The largest clusters in each month consistently scored cultivars higher for taste liking than the overall cohort and monthly averages. This indicates that for most consumers, the taste of rocket is acceptable year-round, with average scores consistently >6.0.

March and April/May had significantly lower taste liking scores than any of the other months. Highest average taste liking was in January, which is contrary to our hypothesis that rocket liking would be greater during summer months. Average scores for July, September, and November were also relatively high (6.0, 6.2, and 6.2, respectively), indicating that in terms of consumer taste liking, spring months show a distinct reduction in acceptability.

Crop cycles of rocket in spring are also typically very long (96.3 days, average) with successive cuts (>2), potentially producing very pungent and bitter leaves. Figure 2 presents consumer perception data of bitterness, hotness, and pepperiness. All these attributes were scored highest in March, with bitterness being a dominant attribute until July. Sweetness perception by comparison remained relatively unchanged, peaking in July. The reason for increased taste liking in January may therefore be explained by the significantly lower perception of hotness of leaves relative to spring and summer months.

#### 3.6.2. Overall Liking

Table 3 presents AHC data and average monthly scores for overall rocket leaf liking. Overall liking encompasses liking of both taste and appearance attributes. Analysis identified three groups for each respective month, except for January, where five clusters were observed. Appearance of leafy salads is known to be a significant factor in consumer purchase intent and liking [2] and the sensory panels determined significantly smaller leaf size and uniformity of shape in January (Appendix A). This disparity between cultivars seems to be compensated for by higher taste liking (Appendix A), suggesting that appearance liking may be secondary to taste liking for some consumers; for example in cluster 4 January (*n* = 43) where scores were all consistently higher than the total cohort average.

The consistency of cultivars was extremely variable during the March and April/May panels. “Fast Grow” (◼-; 6.8) was scored significantly higher on average than all the other samples in March and preferred by all three cluster groups. A similar pattern of inconsistency was observed in the autumn months of September and November. This further suggests that seasonal transitions result in more variable rocket produce.

### 3.7. Principal Component Analysis

#### 3.7.1. Relationships between Consumer Liking and Perceptions

Despite sensory panels not detecting significant differences in sweet or bitter tastes, consumers were able to do so, and this significantly affected their liking for rocket throughout the growing season. This is likely due to the increased diversity of taste receptor profiles present within the population compared with the sensory panel [35].

PCA of the data sets from each consumer panel month revealed a distinct separation between sweetness perception and hotness, bitterness, and pepperiness perceptions along PC1 (Figure 3a). Taste and overall liking are in turn more positively associated with sweetness perception in the upper left quadrant along the PC4 axis, which is in agreement with previous observations in rocket [9]. These data are therefore strong evidence that most consumers are likely to reject rocket if it is too pungent and bitter, as is found in samples received in March and April/May.

#### 3.7.2. Influence of Growing Temperatures on Consumer Preference and Perceptions 

Despite the higher sugar concentrations in July, this does not colocalize with overall/taste liking within the PCA (Figure 3b). The months of March and April/May are in fact most negatively associated with consumer liking, and January and November positively associated with these. Therefore, rocket produced at cooler temperatures is more likely to be preferred by consumers, as bitterness and hotness perceptions are likely to be lower in these months (Figure 3a).

#### 3.7.3. Influence of Cultivation Practice on Consumer Liking and Perceptions

Figure 3b shows that cut number also explains some separation for taste and overall liking. Rocket leaves that were of first cut are generally more common in the upper left quadrant of the PCA plot, and more closely associated with taste liking and sweetness perception (Figure 3). Leaves with more than two cuts separate in the opposite direction towards the lower right quadrant, in the direction of bitter/hotness/pepperiness perception. Anecdotal evidence of traditional cultivation practices by growers has suggested that second cuts (and above) are preferred, because leaves are more uniform, more greatly serrated in shape, and have a more intense flavour. These assertions are in agreement with this study; however none are associated with positive consumer liking or taste liking of rocket leaves. While a subset of consumers may prefer cultivars with increased hotness (as seen in AHC analysis; Appendix A and Table 3) consumers generally do not like this attribute, and prefer milder, sweeter leaves. Thus, conventional agronomic practices of harvesting multiple cuts of rocket may be detrimental to consumer acceptance; particularly in spring months when crop cycles are longer, and the growing season is transitioning.

There is also a significant gap in research more generally about the response of rocket species and cultivars to differences in growth environments and cultivation practices. In this study cultivars were supplied from various growers, and this variable was not controlled so as to assess “real-world” differences in rocket consistency as supplied to consumers. Future studies should aim to assess the variability of multiple cultivars across growing regions, and sample multiple cuts. Such studies are logistically difficult to organise, however it would provide valuable information on how environment influences GSL and hydrolysis product formation on a genotypic basis and how performance of cultivars varies according to the environment. Controlled environment studies have begun to explore these effects [17], but none have to date accounted for variances in soil composition or climatic conditions in the field.

#### 3.7.4. Influence of Glucosinolate Contents on Consumer Liking and Perceptions

Higher concentrations of GER are associated with higher taste and overall liking of rocket leaves (Figure 3a). This trend is in opposition to the abundance of GSV, which is in the bottom right quadrant with bitter/hotness/pepper perceptions. DMB is also separated from GSV and negatively associated with hotness, bitterness, and pepperiness perceptions. This is new evidence that suggests the ratio between monomer and dimer forms of GSV may play a significant role in determining consumer acceptability of rocket leaves. Nothing is known of the genetic mechanisms responsible for the biosynthesis of GSV and DMB, or the mechanisms responsible for determining their relative abundances; but it is generally accepted that SAT (derived from GSV and responsible for pungency) is produced from the monomer form [15]. This may therefore explain the association between GSV and perception traits and indicates that DMB has no objectionable taste of its own.

## 4. Conclusions

This study has found evidence for significant sensorial variability in rocket leaves produced over the course of a growing season as a result of varied cultivation practices and growing locations. This in turn results in variations in consumer liking, which may influence purchase intents and repurchase of rocket leaf products. Seasonal practices, such as growth temperature and the number of cuts crops received, underlie changes in phytochemical composition and may result in the production of overly pungent leaves that consumers are likely to reject. To produce more consistent and acceptable rocket leaves, the practice of multiple harvests should be reserved for developing products targeted at those consumers who like high pungency leaves. First cuts tend to be milder and could therefore be marketed to a wider set of consumers that prefer sweeter leaves and low levels of pungency and bitterness.

## Figures and Tables

**Figure 1 foods-09-01799-f001:**
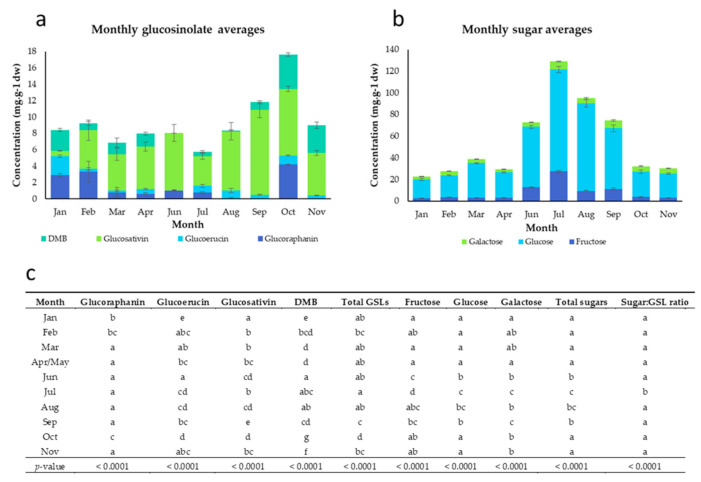
Average glucosinolate concentrations (**a**), sugar concentrations (**b**), and analysis of variance (ANOVA) pairwise comparisons (post-hoc Tukey’s honest significant difference; (**c**) of rocket leaves observed on a monthly basis. Significant differences of glucosinolate concentrations between each sampling month are indicated by differing lower case letters within each column. Concentrations are expressed as mg·g^−1^ of dry weight. Error bars represent standard error of the mean of each compound. See insets for compound colour coding (**a**,**b**). For individual cultivar composition data of samples received each month see Appendix A.

**Figure 2 foods-09-01799-f002:**
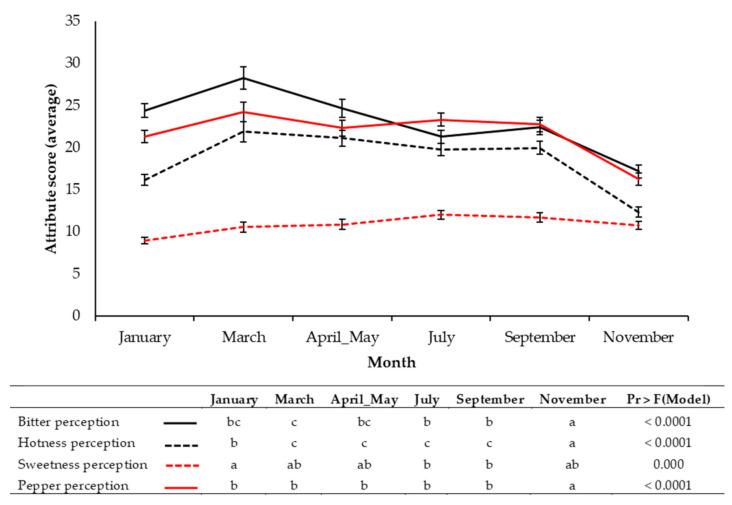
Consumer perceptions of bitter, hotness, sweetness, and pepper attributes of rocket leaves on a bimonthly basis over a growing season. Inset table presents the results of analysis of variance (ANOVA) pairwise comparisons (post-hoc Tukey’s honest significant difference). Significant differences for each perception attribute are indicated by differing lower case letters within rows. See inset for colour coding of attributes. Values are presented as normalized averages of each respective consumer assessment. See Appendix A for the numbers of participants in each respective consumer panel.

**Figure 3 foods-09-01799-f003:**
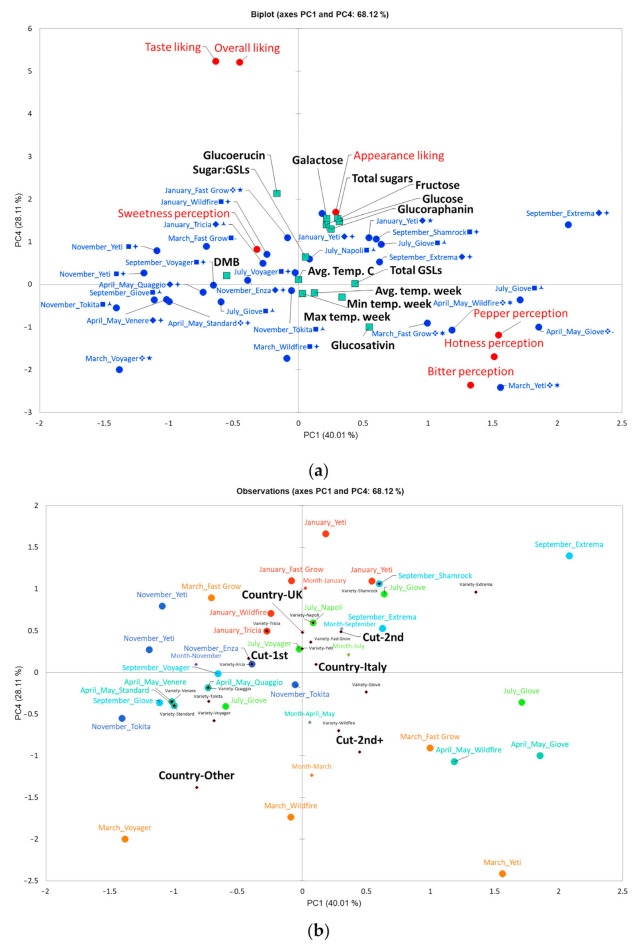
Principal component analysis of consumer liking and perception data of rocket cultivars over the course of one growing season. Biplots display principal components (PCs) 1 and 4, which represent 68.1% of variation within the data; (**a**) factor loadings plot and (**b**) factor scores plot. In (**a**) red circles represent consumer liking and perception attributes, blue circles average monthly samples, and teal squares supplementary phytochemical, and cultivation temperature. Variable label symbols (refer to Table 1): ◼ = 1st cut; ◆ = second cut; ❖ = 2nd + cut; 🟀 = <30 day crop cycle; 🟄 = 31–60 day crop cycle; ★ = 61–90 day crop cycle; ✶ = >91 day crop cycle. In (**b**), see inset for monthly colour coding of samples. Black and red diamonds indicate supplementary variable centroids.

**Table 1 foods-09-01799-t001:** Cultivation and origin details of rocket samples used for sensory and consumer analyses.

Month	Cultivar	Symbol Code	Species	Country of Origin	Growth Environment	Cut Number	Crop Cycle Length (Days)	Dry Matter (%)
January	Yeti	◆🟄	*D. tenuifolia*	Italy	Polytunnel	2nd	60	8.1
	Yeti	◆★	*D. tenuifolia*	Italy	Polytunnel	2nd	71	8.3
	Tricia	◆🟀	*D. tenuifolia*	Italy	Polytunnel	2nd	24	9.5
	Wildfire	◼🟄	*D. tenuifolia*	Italy	Polytunnel	1st	51	8.7
	Fast Grow	❖★	*D. tenuifolia*	Italy	-	2nd+	70	8.1
							**55.2**	**8.5**
February	Venere	❖🟀	*D. tenuifolia*	Italy	Polytunnel	2nd+	27	8.9
	Wildfire	❖🟀	*D. tenuifolia*	Italy	Polytunnel	2nd+	23	13.4
	Selezione Enza	❖✶	*D. tenuifolia*	Italy	Polytunnel	2nd+	124	8.8
	Tricia	❖🟀	*D. tenuifolia*	Italy	Polytunnel	2nd+	24	7.9
	Fast Grow	◆★	*D. tenuifolia*	Italy	-	2nd	85	9.4
							**56.6**	**9.7**
March	Yeti	❖✶	*D. tenuifolia*	Italy	Polytunnel	2nd+	137	-
	Wildfire	◼🟄	*D. tenuifolia*	USA	Open field	1st	31	7.9
	Fast Grow	◼-	*D. tenuifolia*	Italy	Polytunnel	1st	-	-
	Voyager	❖★	*D. tenuifolia*	India	Open field	2nd+	85	-
	Fast Grow	❖✶	*D. tenuifolia*	Italy	-	2nd+	132	-
							**96.3**	-
April	Wildfire	❖✶	*D. tenuifolia*	Italy	Polytunnel	2nd+	180	9.3
	Venere	◆🟄	*D. tenuifolia*	Italy	Polytunnel	2nd	35	7.8
	Standard	❖🟄	*D. tenuifolia*	Spain	Polytunnel	2nd+	40	8.1
	Giove	❖-	*D. tenuifolia*	Italy	Polytunnel	2nd+	-	9.3
	Quaggio	◆🟄	*D. tenuifolia*	Italy	Polytunnel	2nd	43	7.7
							**74.5**	**8.4**
June	Giove	◆🟀	*D. tenuifolia*	Italy	Polytunnel	2nd	30	6.7
	Giove	◼🟀	*D. tenuifolia*	Italy	Polytunnel	1st	23	5.9
	Torino	◼🟄	*D. tenuifolia*	UK	Open field	1st	40	12.3
	Giove	◼🟀	*D. tenuifolia*	Italy	Polytunnel	1st	23	7.5
	Extrema	◆🟄	*D. tenuifolia*	Italy	Polytunnel	2nd	39	6.4
							**31.0**	**7.8**
July	Giove	◼🟀	*D. tenuifolia*	Italy	Polytunnel	1st	20	8.0
	Giove	◼🟀	*D. tenuifolia*	Italy	Polytunnel	1st	20	6.6
	Napoli	◼🟀	*D. tenuifolia*	UK	Open field	1st	24	6.1
	Voyager	◼🟄	*D. tenuifolia*	UK	Open field	1st	44	14.8
	Giove	◆🟀	*D. tenuifolia*	Italy	Polytunnel	2nd	28	7.3
							**27.2**	**8.6**
August	Giove	◼🟀	*D. tenuifolia*	Italy	Polytunnel	1st	24	9.7
	Extrema	◆🟄	*D. tenuifolia*	Italy	Polytunnel	2nd	33	8.5
	Voyager	◼🟀	*D. tenuifolia*	UK	Glasshouse	1st	24	6.6
							**27.0**	**8.3**
September	Voyager	◼🟄	*D. tenuifolia*	UK	Glasshouse	1st	35	8.7
	Shamrock	◼🟄	*D. tenuifolia*	UK	Open field	1st	38	9.6
	Extrema	◆🟄	*D. tenuifolia*	Italy	Polytunnel	2nd	31	7.0
	Extrema	◆🟄	*D. tenuifolia*	Italy	Polytunnel	2nd	31	9.7
	Giove	◼🟀	*D. tenuifolia*	Italy	Polytunnel	1st	22	5.7
							**31.4**	**8.1**
October	Selezione Enza	◼🟀	*D. tenuifolia*	Italy	Polytunnel	1st	25	5.8
	Selezione Enza	◆🟄	*D. tenuifolia*	Italy	Polytunnel	2nd	33	6.5
	Venere	◼🟀	*D. tenuifolia*	Italy	Polytunnel	1st	23	6.3
	Multi	◆🟀	*D. tenuifolia*	Italy	Polytunnel	2nd	30	6.2
	Napoli	◆-	*D. tenuifolia*	UK	Glasshouse	2nd	-	8.1
							**27.8**	**6.6**
November	Yeti	◼🟄	*D. tenuifolia*	Italy	Polytunnel	1st	31	6.3
	Yeti	◼🟄	*D. tenuifolia*	Italy	Polytunnel	1st	32	6.5
	Selezione Enza	◆🟄	*D. tenuifolia*	Italy	Polytunnel	2nd	60	6.4
	Tokita	◼🟀	*E. sativa*	Italy	Polytunnel	1st	24	10.4
	Tokita	◼🟀	*E. sativa*	Italy	Polytunnel	1st	24	7.2
							**34.2**	**7.4**

◼ = 1st cut; ◆ = second cut; ❖ = 2nd+ cut; 🟀 = <30 day crop cycle; 🟄 = 31–60 day crop cycle; ★ = 61–90 day crop cycle; ✶ = >91 day crop cycle. Symbol codes denoting cut number and crop cycle length are also utilised in subsequent text tables and figures. Numbers in bold are monthly averages. N.B. Hyphens (-) indicate data was not supplied from the grower.

**Table 2 foods-09-01799-t002:** Analysis of variance (ANOVA), Tukey’s honest significant difference (HSD), and pairwise comparison data for sensory attributes of rocket salad leaves supplied to the United Kingdom over the course of one year.

Attribute	Month										
	January	February	March	April	June	July	August	September	October	November	*p*-Value
*Appearance*											
Leaf size	a	ab	ab	b	b	ab	b	ab	b	ab	0.005
Uniformity of size	a	ab	b	b	b	b	b	b	b	b	<0.0001
*Odour*											
Green	b	ab	a	ab	a	a	ab	ab	ab	a	0.013
Stalky	bc	abc	bc	ab	a	ab	ab	c	abc	ab	<0.0001
Earthy	c	ab	ab	abc	a	abc	ab	ab	bc	abc	<0.0001
Peppery	b	a	a	a	a	a	ab	a	a	a	<0.0001
Sweet	a	a	ab	ab	ab	b	b	ab	ab	b	<0.0001
Mustard	a	a	a	a	a	a	a	a	a	a	0.029
*Mouthfeel*											
Crisp	a	abc	bc	c	c	c	c	c	c	ab	<0.0001
Drying	ab	ab	a	a	a	ab	ab	b	b	a	<0.0001
*Taste*											
Sour	c	abc	a	bc	ab	abc	c	abc	abc	a	<0.0001
Savoury	d	ab	bcd	abcd	a	abc	cd	bcd	abcd	d	<0.0001
*Flavour*											
Stalky	ab	a	ab	ab	a	ab	a	b	ab	a	0.001
Peppery	b	ab	a	a	a	a	ab	ab	a	a	0.001
Earthy	c	ab	abc	abc	a	bc	abc	abc	abc	abc	0.005
*Aftereffects*											
Sweet	ab	a	ab	ab	ab	b	ab	ab	ab	ab	0.024
Sour	b	ab	a	ab	ab	ab	ab	ab	ab	a	0.006
Savoury	cd	ab	cd	ab	a	abc	bcd	bcd	bcd	d	<0.0001
Peppery	b	ab	ab	ab	ab	ab	ab	ab	ab	a	0.014
Green	ab	a	ab	a	a	ab	ab	ab	b	ab	0.008
Earthy	a	a	a	a	a	a	a	a	a	a	0.031

Different letters within each row indicate significant differences between attributes in each sampling month (a = the lowest level of significant difference). Refer to Appendix A for data and standard errors.

**Table 3 foods-09-01799-t003:** Overall liking of rocket cultivars for the clusters of consumers obtained from agglomerative hierarchical clustering.

Month	Cluster	Cultivar					Average
		Tricia ◆🟀	Yeti ◆🟄	Wildfire ◼🟄	Fast Grow ❖★	Yeti ◆★	
January	1 (*n* = 15)	3.9	5.8	4.1	4.0	5.3	4.6
	2 (*n* = 6)	7.5	6.8	2.8	7.8	7.2	6.4
	3 (*n* = 18)	6.2	3.6	6.9	6.6	6.5	6.0
	4 (*n* = 43)	7.6	7.0	7.3	6.7	7.0	7.1
	5 (*n* = 19)	4.8	7.2	6.2	6.5	6.9	6.3
	All	6.3 ns	6.7 ns	6.3 ns	6.3 ns	6.3 ns	6.4 D
March		Yeti ❖✶	Wildfire ◼🟄	Fast Grow ◼-	Fast Grow ❖✶	Voyager ❖★	Average
	1 (*n* = 29)	6.8	6.3	7.0	6.3	5.3	6.3
	2 (*n* = 20)	2.8	4.6	7.3	5.7	5.1	5.1
	3 (*n* = 6)	4.0	2.2	4.2	2.7	3.8	3.4
	All	5.0 a	5.2 a	6.8 b	5.7 a	5.1 a	5.6 A
April/May		Venere ◆🟄	Wildfire ❖✶	Standard ❖🟄	Giove ❖-	Quaggio ◆🟄	Average
	1 (*n* = 10)	5.3	5.1	6.0	4.6	2.0	4.6
	2 (*n* = 39)	6.7	4.2	6.1	3.6	6.8	5.5
	3 (*n* = 41)	5.9	6.1	5.7	7.0	7.1	6.4
	All	6.2 b	5.2 a	5.9 ab	5.3 a	6.4 b	5.8 AB
July		Giove ◼🟀	Giove ◼🟀	Giove ◼🟀	Napoli ◼🟀	Voyager ◼🟄	Average
	1 (*n* = 43)	6.2	6.2	3.9	5.4	5.2	5.4
	2 (*n* = 12)	3.5	3.8	6.3	6.0	5.3	5.0
	3 (*n* = 45)	6.2	6.6	7.2	6.8	7.4	6.8
	All	5.9 ns	6.1 ns	5.7 ns	6.1 ns	6.2 ns	6.0 BC
September		Voyager ◼🟄	Shamrock ◼🟄	Extrema ◆🟄	Extrema ◆🟄	Giove ◼🟀	Average
	1 (*n* = 65)	6.8	7.0	6.9	7.2	6.1	6.8
	2 (*n* = 9)	4.6	4.8	4.0	6.6	2.6	4.5
	3 (*n* = 15)	5.3	5.9	4.6	3.3	6.6	5.1
	All	6.3 ab	6.6 b	6.2 ab	6.4 ab	5.8 a	6.3 CD
November		Yeti ◼🟄	Yeti ◼🟄	Selezione Enza ◆🟄	Tokita ◼🟀	Tokita ◼🟀	Average
	1 (*n* = 40)	7.2	6.8	7.3	6.6	7.4	7.1
	2 (*n* = 19)	5.2	5.1	6.3	4.3	4.1	5.0
	3 (*n* = 27)	7.0	6.4	3.7	6.0	6.0	5.8
	All	6.7 b	6.3 ab	5.9 a	5.9 a	6.2 ab	6.2 CD

Letters indicate ANOVA pairwise comparison significances (Tukey’s HSD). Lower case letters within the “All” rows of each month refer to individual cultivar scores across each consumer panel month. Upper case letters in the “Average” column refer to significant differences in monthly average scores. Where letters are different a significant difference at the *p =* ≤0.0001 level was observed. ◼ = 1st cut; ◆ = second cut; ❖ = 2nd + cut; 🟀 = <30 day crop cycle; 🟄 = 31–60 day crop cycle; ★ = 61–90 day crop cycle; ✶ = >91 day crop cycle; ns = no significant difference. N.B. Hyphens (-) indicate data was not supplied from the grower.

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
