# Peer review of "High Glucosinolate Content in Rocket Leaves (Diplotaxis tenuifolia and Eruca sativa) after Multiple Harvests Is Associated with Increased Bitterness, Pungency, and Reduced Consumer Liking"

_foods, 2020, doi:10.3390/foods9121799_

Round 1

Reviewer 1 Report

The work reported by Bell et al. establishes a correlation between high glucosinolate content in rocket leaves, cultivation practices (multiple harvests) and increased bitterness, pungency, and consequent reduced consumer’s acceptance. This is an interesting work, but I feel that the authors should improve it by focusing on the essential correlation they report. Overall, the manuscript is too long and discuss with too much detail certain results. This makes the manuscript difficult to follow. My suggestion is to present some data as supplementary material. Figure 1, for instance, although important, can be just discussed in the manuscript, providing it as Supplementary Figure 1. Also, the results in Figure 3 are consolidated in Table 2, so Figure 3 could be presented as Supplementary Figure 2.

Table 3 is another table that can be provided as Supplementary material. In fact, I think the topic 3.6 should be merged with 2.4 and presented in that section (Materials and Methods).  

Authors considered two subsections in point 3.7., “Liking of taste” and “Overall liking” but fail to explain the need of having this differentiation. I suggest simplifying the discussion of the “Consumer acceptability and perception”, briefly discussing “liking of taste” and focusing the discussion in the “Overall liking”. Accordingly, at least Table 4 should be presented as supplementary material.  

Line 104: “As such, the cultivars tested each month were not always the same.” Although I can understand the reasons why the authors followed this strategy, I think that it would be also important to have a parallel study using the same cultivars. This way, the effect of the genetic background on the quality of rocket leaves, particularly in what concerns to the correlation between GSLs accumulation, cultivation practices and consumer´s acceptance, could be discussed more accurately. Certainly that some cultivars may have lower endogenous levels of GSLs, while others are more tolerant to the cultivation practices, and this point should be discussed by the authors in the manuscript.

In Figure 1 and throughout the manuscript, the authors used “Significant differences are indicated by differing lower case letters” - this is not very clear and should be clarified.

Table 1: the symbols that are used to identify the number of cuts and crop cycles length should be placed in the respective columns in this Table to be clear in the next tables what they mean and why seems to be a repletion on Table 1.

Figure 5: delete repeated “Figure 5”

Author Response

- The work reported by Bell et al. establishes a correlation between high glucosinolate content in rocket leaves, cultivation practices (multiple harvests) and increased bitterness, pungency, and consequent reduced consumer’s acceptance. This is an interesting work, but I feel that the authors should improve it by focusing on the essential correlation they report. Overall, the manuscript is too long and discuss with too much detail certain results. This makes the manuscript difficult to follow. My suggestion is to present some data as supplementary material. Figure 1, for instance, although important, can be just discussed in the manuscript, providing it as Supplementary Figure 1. Also, the results in Figure 3 are consolidated in Table 2, so Figure 3 could be presented as Supplementary Figure 2.

Response: Many thanks for your evaluation of our work. We appreciate the time and effort you have taken to help us improve the manuscript. We have taken these comments on board and made efforts to reduce its length (please see track changes). Figure 1 has been made into ‘Supplementary Figure S1’ and Figure 3 is now ‘Supplementary Figure S2’, as per your recommendation.

- Table 3 is another table that can be provided as Supplementary material. In fact, I think the topic 3.6 should be merged with 2.4 and presented in that section (Materials and Methods). 

Response: Table 3 is now ‘Supplementary Table S1’ as suggested. Section 3.6 has also been moved and combined into section 2.4 as suggested.

- Authors considered two subsections in point 3.7., “Liking of taste” and “Overall liking” but fail to explain the need of having this differentiation. I suggest simplifying the discussion of the “Consumer acceptability and perception”, briefly discussing “liking of taste” and focusing the discussion in the “Overall liking”. Accordingly, at least Table 4 should be presented as supplementary material. 

Response: A clarification of liking and overall liking has been inserted into sections 3.7.1 and 3.7.2. These respective sections have been shortened, and Table 4 is now ‘Supplementary Table S2’.

- Line 104: “As such, the cultivars tested each month were not always the same.” Although I can understand the reasons why the authors followed this strategy, I think that it would be also important to have a parallel study using the same cultivars. This way, the effect of the genetic background on the quality of rocket leaves, particularly in what concerns to the correlation between GSLs accumulation, cultivation practices and consumer´s acceptance, could be discussed more accurately. Certainly that some cultivars may have lower endogenous levels of GSLs, while others are more tolerant to the cultivation practices, and this point should be discussed by the authors in the manuscript.

Response: Thank you for raising this point and we agree. While the study of cultivar consistency between growth environments was beyond the scope of this study we have inserted additional discussion to section 3.8.3. that explains the need for further research in this area.

- In Figure 1 and throughout the manuscript, the authors used “Significant differences are indicated by differing lower case letters” - this is not very clear and should be clarified.

Response: The Figure and Table legends have been clarified throughout according to this suggestion.

- Table 1: the symbols that are used to identify the number of cuts and crop cycles length should be placed in the respective columns in this Table to be clear in the next tables what they mean and why seems to be a repletion on Table 1.

Response: Symbols used in Table 1 have been entered into their own column to improve visibility. A statement in the table footnote now explains their relevance and connection to Table 3 and Figure 3.

- Figure 5: delete repeated “Figure 5”

Response: Thank you for spotting this error, it has been corrected.

Reviewer 2 Report

The topic of the article is quite interesting since involves a vegetable crop which is increasingly consumed for its beneficial healthy related properties. As clearly stated in previous publications, a number of them written by the same authors, research involving rocks is growing but still lacking: “Very few studies have been conducted on sensory aspects (Bell et al., 2017, Bell et al., 2017, D’Antuono et al., 2009, Lokke et al., 2012, Pasini et al., 2011), and only one on consumer preferences and perceptions (Bell, Methven, & Wagstaff, 2017). Similarly, only a single study has explored the supply chain of rocket in a commercial environment (Bell, Yahya et al., 2017).” from Bell L et al., Rocket science: A review of phytochemical & health-related research in Eruca & Diplotaxis species.Food Chemistry: X Volume 1, 30 March 2019, 100002.

The authors have clear expertise in the sector as attested by the numerous publication on this topic, anyway I found that some weak points need to be cleared before publication:
1) in this article a specific consumer’s panel has been used (University students and workers), therefore, the results are not general for “the consumers” but reflect the specific consumer's panel. One of the main points of this article is to understand the “Influence of agronomic practice on consumer liking and perceptions “ (line 544) anyway, the results shown are specifically linked to the panel employed which is not representative of the whole rocket consumers UK population. Indeed, some outcomes like Lines 553-556, the authors found that “While a subset of consumers may prefer cultivars with increased hotness […] consumers generally do not like this attribute […]Thus, conventional agronomic practices of harvesting multiple cuts of rockets may actually be detrimental to consumer acceptance”, are not coherent indeed as the authors themselves pointed out in a previous article, whether hotness is a positive or negative character in rockets depends on the consumer's group:” This suggests hotness is preferable for one group of consumers but is rejected by another.” Bell, Methven, Wagstaff, The influence of phytochemical composition and resulting sensory attributes on preference for salad rocket (Eruca sativa) accessions by consumers of varying TAS2R38 diplotype. Food Chemistry. Volume 222, 1 May 2017, Pages 6-17 Please, underline that the results of this article aren’t absolute ones but depend on the specific consumer's panel.
2) samples provided by one food manufacturing company (not the main or the only one which provides arugula in the UK);
3) not enough information regarding the agronomic practices (see Table 5: data were not always “supplied from the grower”.)moreover, since arugula is imported from different countries in which different agronomic chemical products are allowed, the factors influencing the chemical compositions are variable and not taken into consideration in this work. Even if the authors stated that they compared cultivation procedures they actually haven’t enough information about each the cultivation practice performed.
4)two different rocket species have been investigated which would have had a different chemical composition even if grown in the same geographical area, during the same season, with the same agronomical practices. There are strong differences between Diplotaxis tenuifolia and Eruca sativa, as the authors themselves pointed out in previous articles “E. sativa is a morphologically diverse species, and some cultivars share similar characteristics to D. tenuifolia” (Bell & Wagstaff, 2014). which was confirmed by other publications. [Kadri Bozokalfa, M., EĹźiyok, D., Ä°lbi, H., Kavak, S., & Kaygısız AĹźçıoÄźul, T. (2011). Evaluation of phenotypic diversity and geographical variation of cultivated (Eruca sativa L.) and wild (Diplotaxis tenuifolia L.) rocket plant. Plant Genetic Resources, 9(3), 454-463. doi:10.1017/S1479262111000657] and lines 63-64:” a significant plant genotypic component determining the pungency of leaves” Indeed, Diplotaxis tenuifolia and Eruca sativa, have a different chemical composition.
Therefore, since the article is quite interesting I suggest the authors modify the article underlining that their outcome is just a preliminary study which needs further analysis in the subsequent years, from a different supply chain/food manufacturing provider, with a sensory analysis performed on a larger consumers’ panel in order to confirm or not these which are, again, just preliminary findings.
I suggest a modification of title (something like: “Preliminary results on rocket leaves (Diplotaxis tenuifolia and Eruca sativa): high glucosinolate content after multiple harvests is associated with increased bitterness, pungency, which affects consumer liking”) abstract and conclusion and a coherent revision of the text.
Other comments:
Line 16 Please replace “determined” with “evaluated” or something similar.
Line 20 “consumer acceptance for the panel test employed”
Line 115 Therefore, you obtained the temperature for each cultivation site and then calculated a mean value for the month. Figure 1 “presents the average temperature data supplied by growers from each farm location” here it looks like the temperature was measured in only one location each month. Or maybe you calculated a mean for the temperature of different cultivation sites in different countries within the same month? Can you please explain this point?
Line 123 How did you trained the panel? Or maybe it was used an already trained panel? Can you insert the gender ratio and the age range of the panel?
Line 134 Consumers: how many samples (number of leaves) were provided to each consumer? Were they able to re-taste the same sample? In some months samples came from different Countries, did the same consumer tasted the samples coming from different production sites? As I pointed out before, samples produced in different countries, even if marketed in the UK in the same month are expected to be different in chemical composition, especially considering the month of November in which two different rocket species have been pooled and analysed.
Line 172 Are ANOVA and Tukey’s test the correct choice for your data? Can you please insert some information about their normality and orthogonality which will make them compatible with parametric tests?
Line 175 It is not clear to me why the post hoc test is “protected”, can you please clarify this point to me?
Lines 195-197 could you add some references to support this statement? Actually, you analyzed the arugula that your chosen supplier provided you which is not the only one in the whole UK.
Line 200 “is almost entirely”: which means that you cannot assure that the arugula produced in Italy was only cultivated in polytunnel. Which, in turn, makes questionable the knowledge of the whole agronomical practices followed.
Lines 251-262 In the months of July and October authors analysed arugula samples from Italy and the UK. I suppose that during these months temperatures were quite different in the two countries (maybe not in summer since samples came from north and east rn Italy, but definitely different in October when samples came from southern regions see lines 45-47) therefore, if there has been a difference linked to the temperature (cooler in the UK I suppose) it was not taken into consideration. I think the authors should have analysed the UK and Italy samples separately. (since the temperature was not significantly different in each country but comparing them I think there were some significant differences). Same goes for all the other samples pooled from different locations: it is not possible to consider a mean temperature since it was different in different countries even during the same months.
Line 523 Please replace “collated” with “collected”
Line 549 “Anecdotal evidence” is based on casual observations rather than proper scientific research or statistics outcomes, please insert a reference to support this statement.

Author Response

1) in this article a specific consumer’s panel has been used (University students and workers), therefore, the results are not general for “the consumers” but reflect the specific consumer's panel. One of the main points of this article is to understand the “Influence of agronomic practice on consumer liking and perceptions “ (line 544) anyway, the results shown are specifically linked to the panel employed which is not representative of the whole rocket consumers UK population. Indeed, some outcomes like Lines 553-556, the authors found that “While a subset of consumers may prefer cultivars with increased hotness […] consumers generally do not like this attribute […]Thus, conventional agronomic practices of harvesting multiple cuts of rockets may actually be detrimental to consumer acceptance”, are not coherent indeed as the authors themselves pointed out in a previous article, whether hotness is a positive or negative character in rockets depends on the consumer's group:” This suggests hotness is preferable for one group of consumers but is rejected by another.” Bell, Methven, Wagstaff, The influence of phytochemical composition and resulting sensory attributes on preference for salad rocket (Eruca sativa) accessions by consumers of varying TAS2R38 diplotype. Food Chemistry. Volume 222, 1 May 2017, Pages 6-17 Please, underline that the results of this article aren’t absolute ones but depend on the specific consumer's panel.

Response: Thank you for this assessment, and we generally agree with the points that have been raised. This issue is an inherent limitation of any consumer study, as many responses may be subjective. We would highlight that very few other studies of horticultural crops have utilised consumer groups over such a long period of time to assess a product. We acknowledge that the pool of consumers that we have utilised may not necessarily be representative of the whole UK population, but the frequency and nature of the study necessitated a close proximity to the University in order to conduct the research. We have inserted a statement within the text that addresses this point (section 2.4).

2) samples provided by one food manufacturing company (not the main or the only one which provides arugula in the UK)

Response: The supplier (Bakkavor) is one of the largest suppliers of rocket leaves to the UK market. The research was partnered with them and made use of their supply chain network in order to obtain detailed information about the source of rocket leaves and have direct access to growers. The company supplies various supermarket chains in the UK and the source of their produce would be similar to other companies providing rocket leaves to the UK market in a typical year (e.g. both Italy and UK grown). We do not see this as a limitation, as without this industry collaboration it would not have been possible to conduct the research at all. Similarly we would point out that no previous studies have been published in this crop (or for many other horticultural crops, for that matter) that encompass the whole supply chain and sensory-consumer responses in this way.

3) not enough information regarding the agronomic practices (see Table 5: data were not always “supplied from the grower”.)moreover, since arugula is imported from different countries in which different agronomic chemical products are allowed, the factors influencing the chemical compositions are variable and not taken into consideration in this work. Even if the authors stated that they compared cultivation procedures they actually haven’t enough information about each the cultivation practice performed.

Response: We wished to determine the variability of rocket produce throughout a growing season and this encompassed all possible differences that might arise according to fertilizer inputs and land management practices etc. The aim was to determine the impacts this inherent variability has on sensory properties and consumer acceptance. It is clear from our data that number of cuts has a significant impact on glucosinolate accumulation, and that the time of year impacts sugar concentrations. The study of additional agronomic parameters and collection of such data would have been impractical, and beyond the scope of our investigation. Future research can now build upon the data we present here to determine the impacts of such interacting factors.

4)two different rocket species have been investigated which would have had a different chemical composition even if grown in the same geographical area, during the same season, with the same agronomical practices. There are strong differences between Diplotaxis tenuifolia and Eruca sativa, as the authors themselves pointed out in previous articles “E. sativa is a morphologically diverse species, and some cultivars share similar characteristics to D. tenuifolia” (Bell & Wagstaff, 2014). which was confirmed by other publications. [Kadri Bozokalfa, M., EĹźiyok, D., Ä°lbi, H., Kavak, S., & Kaygısız AĹźçıoÄźul, T. (2011). Evaluation of phenotypic diversity and geographical variation of cultivated (Eruca sativa L.) and wild (Diplotaxis tenuifolia L.) rocket plant. Plant Genetic Resources, 9(3), 454-463. doi:10.1017/S1479262111000657] and lines 63-64:” a significant plant genotypic component determining the pungency of leaves” Indeed, Diplotaxis tenuifolia and Eruca sativa, have a different chemical composition.

Response: Rocket is a generalised term for several species that are marketed under the same name. We included samples of Eruca sativa in the study as it forms part of the supply of rocket leaves to the UK within a typical growing season, as it is noted for its superior establishment characteristics compared to Diplotaxis tenuifolia at times of the season when supply is limited (e.g. in winter). Also, processors, supermarkets and consumers do not distinguish between rocket species when selling/consuming leaves, and so there would be no point in us doing so either. We acknowledge that the overall phytochemical profiles of E. sativa and D. tenuifolia are not the same, however their specific glucosinolate and sugar profiles are, and it was these specific metabolites that were the focus of this study. Indeed previous research has found that it is not possible to distinguish the two species on the basis of glucosinolate profile alone (see Pasini et al. 2012; Food Chemistry, 133(3) p1025-1033).

Therefore, since the article is quite interesting I suggest the authors modify the article underlining that their outcome is just a preliminary study which needs further analysis in the subsequent years, from a different supply chain/food manufacturing provider, with a sensory analysis performed on a larger consumers’ panel in order to confirm or not these which are, again, just preliminary findings.
I suggest a modification of title (something like: “Preliminary results on rocket leaves (Diplotaxis tenuifolia and Eruca sativa): high glucosinolate content after multiple harvests is associated with increased bitterness, pungency, which affects consumer liking”) abstract and conclusion and a coherent revision of the text.

Response: We would like to thank the reviewer for this suggestion, however we do not feel that these data are preliminary in nature. We have published sensory, consumer, and supply chain studies on rocket (and other species) in recent years and this paper builds upon and confirms many of the hypotheses we have previously formed. All studies and data are preliminary insofar as more data can always be collected by subsequent studies to confirm/disprove the observations made therein. The trends and observations we have made are statistically robust and all analyses were conducted with appropriate replication. 

Line 16 Please replace “determined” with “evaluated” or something similar.

Response: The word has been changed.

Line 20 “consumer acceptance for the panel test employed”

Response: This sentence has been modified.

Line 115 Therefore, you obtained the temperature for each cultivation site and then calculated a mean value for the month. Figure 1 “presents the average temperature data supplied by growers from each farm location” here it looks like the temperature was measured in only one location each month. Or maybe you calculated a mean for the temperature of different cultivation sites in different countries within the same month? Can you please explain this point?

Response: Yes, thank you. The temperatures presented are averages of all cultivation sites giving a representative picture for the time of year. Additional text has been inserted to make this clearer (section 3.2).

Line 123 How did you trained the panel? Or maybe it was used an already trained panel? Can you insert the gender ratio and the age range of the panel?

Response: The gender ratio of the panel has been inserted but their age-related data are unavailable. Additional text about the training of the panel has been inserted.

Line 134 Consumers: how many samples (number of leaves) were provided to each consumer? Were they able to re-taste the same sample? In some months samples came from different Countries, did the same consumer tasted the samples coming from different production sites? As I pointed out before, samples produced in different countries, even if marketed in the UK in the same month are expected to be different in chemical composition, especially considering the month of November in which two different rocket species have been pooled and analysed.

Response: The information relating to number of leaves and re-tasting has been entered into section 2.4. We asked participants to attend as many of the study visits as possible within the year. A total of 55 out of 101 volunteer consumers attended every session. Despite not all consumers being able to visit every session, this provided a high level of consistency between the sample sessions each month. This information has now been incorporated into section 2.4.

Line 172 Are ANOVA and Tukey’s test the correct choice for your data? Can you please insert some information about their normality and orthogonality which will make them compatible with parametric tests?

Response: This information has now been entered into section 2.6.2. ANOVA was an appropriate test for the data and is used routinely in sensory and consumer study evaluations.

Line 175 It is not clear to me why the post hoc test is “protected”, can you please clarify this point to me?

Response: The analyses were protected to ensure that no false positive results (type I errors) were included in our interpretations of the data.

Lines 195-197 could you add some references to support this statement? Actually, you analyzed the arugula that your chosen supplier provided you which is not the only one in the whole UK.

Response: A reference has been added in support of this statement. It would have been impractical to obtain rocket samples from every rocket supplier to the UK for an entire year. The rocket that was used in this study is representative of typical produce supplied to the UK and eaten by UK consumers.

Line 200 “is almost entirely”: which means that you cannot assure that the arugula produced in Italy was only cultivated in polytunnel. Which, in turn, makes questionable the knowledge of the whole agronomical practices followed.

Response: This sentence has been modified. As previously stated, we did not attempt to account for every agronomic practice followed by growers, but did ask specifically if their crops were grown in open field, polytunnel, or glasshouse. This information was provided by all except two growers throughout the growing season (see Table 1).

Lines 251-262 In the months of July and October authors analysed arugula samples from Italy and the UK. I suppose that during these months temperatures were quite different in the two countries (maybe not in summer since samples came from north and east rn Italy, but definitely different in October when samples came from southern regions see lines 45-47) therefore, if there has been a difference linked to the temperature (cooler in the UK I suppose) it was not taken into consideration. I think the authors should have analysed the UK and Italy samples separately. (since the temperature was not significantly different in each country but comparing them I think there were some significant differences). Same goes for all the other samples pooled from different locations: it is not possible to consider a mean temperature since it was different in different countries even during the same months.

Response: Respectfully, this is accounted for in the statistical analysis of the temperature data supplied to us by growers. It was presented in Figure 1 (now Supplementary Figure S1). ANOVA was used to assess the variance of temperature data between sampling months and found a high level of significant difference (P<0.0001) for average growth temperatures across the year. This is also signified and accounted for in the presentation of standard error bars within Figure S1. Moreover, the hypothesis we were testing was not to test the differences between countries, but the differences in sensory/chemical properties of rocket leaves between months of the year.

Line 523 Please replace “collated” with “collected”

Response: The sentence has been amended.

Line 549 “Anecdotal evidence” is based on casual observations rather than proper scientific research or statistics outcomes, please insert a reference to support this statement.

Response: The point being raised in the sentence is exactly that; growers base their practices on informal observations rather than formal scientific analysis. The point we were making here is that practices such as multiple harvests may improve visual quality but are actually detrimental to rocket sensory quality and consumer acceptance.